# EGFR-Targeted Antibody–Drug Conjugate to Different Aminobisphosphonates: Direct and Indirect Antitumor Effects on Colorectal Carcinoma Cells

**DOI:** 10.3390/cancers16071256

**Published:** 2024-03-22

**Authors:** Leila Pisheh, Serena Matis, Martina Taglieri, Linda Di Gregorio, Roberto Benelli, Alessandro Poggi

**Affiliations:** 1Molecular Oncology and Angiogenesis Unit, IRCCS Ospedale Policlinico San Martino, 16132 Genova, Italy; l.pisheh@erasmusmc.nl (L.P.); serena.matis@hsanmartino.it (S.M.); martina.taglieri@hsanmartino.it (M.T.); linda.digregorio@hsanmartino.it (L.D.G.); roberto.benelli@hsanmartino.it (R.B.); 2Department of Pulmonary Medicine, Erasmus Medical Center, 3015 Rotterdam, The Netherlands

**Keywords:** gamma delta T lymphocyte, antibody–drug conjugate, colorectal cancer, aminobisphosphonate, epidermal growth factor receptor

## Abstract

**Simple Summary:**

Colorectal carcinoma (CRC) is a prevalent form of cancer globally. Despite advancements in diagnosing and treating CRC patients, the current therapies and control measures are insufficient in improving outcomes. This study aims to assess the potential effectiveness of immunotherapy utilizing γδ T cells for CRC. To accomplish this, conventional in vitro cultures and spheroids, a practical 3D culture system, were employed as a reliable in vitro model to examine the characteristics and behavior of CRC tumor cells. New antibody–drug conjugates (ADCs) were developed to specifically target EGFR^+^ CRC cells and activate the Vδ2-T-cell-mediated response by delivering an aminobisphosphonate, ultimately leading to the killing of CRC cells. The efficacy and cytotoxicity of these ADCs were assessed to determine the relevance of this conjugation approach for eliminating tumor cells.

**Abstract:**

Antibody––drug conjugates (ADCs) are a promising delivery system that involves linking a monoclonal antibody (mAb) to a specific drug, such as a cytotoxic agent, to target tumor cells. This new class of antitumor therapy acts as a “biological missile” that can destroy tumor cells while increasing the therapeutic index and decreasing toxicity. One of the most critical factors in ADC design is selecting a target antigen that is highly expressed on the surface of cancer cells. In this study, we conjugated Cetuximab (Cet), a monoclonal antibody that targets the epidermal growth factor receptor (EGFR), to aminobisphosphonates (N-BPs) such as ibandronate (IBA) or risedronate (RIS) or zoledronate (ZA). Cetuximab is administered to patients with metastatic colorectal carcinoma (mCRC) with a wild-type (WT) EGFR transduction pathway. Also, it is well established that N-BPs can trigger the antitumor activity of Vδ2 T cells in both in vitro and in vivo experimental models. The resulting ADCs were added in co-culture to assess the effect on CRC cell line proliferation and sensitivity to Vδ2 T antitumor lymphocytes in comparison with the native antibody. These assays have been performed both in conventional and 3D spheroid cultures. We found that all three ADCs can increase the inhibitory effect on cell proliferation of the WT-EGFR cell line Caco-2 while only Cet-RIS and Cet-ZA can increase the cytotoxicity mediated by Vδ2 T cells against both WT and EGFR-mutated CRC cell lines (Caco-2, DLD-1, and HCT-116). Also, the ADCs can trigger the cell proliferation of Vδ2 T cells present in peripheral blood and tumor specimens. Our findings indicate that anti-EGFR antibodies bound to N-BPs can improve the antitumor effects of the native antibody possibly increasing the therapeutic effect.

## 1. Introduction

Antibody–drug conjugates (ADCs) are an emerging class of combination therapy that involves coupling a monoclonal antibody to a drug (or “payload”), such as a toxic agent, immune stimulatory cytokines, or an antimitotic agent, to target antigens highly expressed in tumor cells and kill them [1,2]. This strategy can deliver the payload to tumor cells while reducing toxicity to healthy tissue, thereby boosting bioavailability and optimizing pharmacokinetic/pharmacodynamic characteristics [3,4,5,6]. The three main components of these therapeutic entities are monoclonal antibodies, drugs, and cleavable or non-cleavable linkers [7,8,9,10]. By the end of 2021, fourteen ADCs had received U.S. Food and Drug Administration (FDA) approval, and more than 100 ADCs were in various stages of clinical development worldwide [11,12].

The most critical factor determining an ADC’s antitumor efficacy and tolerability is target selection. The target antigen should preferably be highly expressed on the surface of cancer cells, making it accessible to the ADC [13]. The rate of internalization is also critical since the antigen-ADC complex is subject to internalization after binding the ADC [8,14]. Epidermal growth factor receptor (EGFR) is a prime target in therapeutic development since numerous malignancies, including colorectal carcinoma (CRC) [15,16], non-small cell lung cancer (NSCLC) [17,18], and head and neck squamous cell carcinoma (HNSCC) [19,20], have abnormal overexpression of EGFR. The FDA-approved anti-EGFR monoclonal antibody cetuximab (Cet) is used as first-line therapy, in combination with chemotherapy, for metastatic, KRAS wild-type CRC [21,22]. While Cet mainly acts blocking EGFR, inhibiting cell proliferation, it may also exhibit antitumor effects by antibody-dependent cell-mediated cytotoxicity (ADCC), engaging distinct Fc receptors on natural killer cells (NKs), γδ T lymphocytes, monocytes, and granulocytes [23].

Recent studies have shown that nitrogen-containing bisphosphonates (N-BPs), such as zoledronate (ZA), risedronate (RIS), and ibandronate (IBA), can specifically boost the growth and antitumor activity of a particular subset of human γδ T cells, known as Vγ9Vδ2 T cells, enhancing their capacity to produce antitumor factors such as IFNγ [24,25]. N-BPs enter monocytes, macrophages, the endothelium, and tumor cells to mediate their anti-cancer activity by blocking farnesyl pyrophosphate synthase (FPPS), a key enzyme in the mevalonate pathway involved in cholesterol synthesis, inducing phosphate antigens (pAg) such as isopentenyl pyrophosphate (IPP) accumulation and triggering Vδ2 T cells [24,25,26,27]. ZA is the most potent N-BP for the therapeutic expansion of Vδ2 T cells [28]. The TCR of Vδ2 cells recognizes pAg bound to the cytoplasmic tail of butyrophilin 3A1 (BTN3A1), a B7-family-related Ig-superfamily protein. This binding causes BTN3A1 to alter its conformation, being detected by Vδ2 and stimulating cell growth and expansion [29].

Recently, we reported the generation of the ADC composed of cetuximab covalently linked to zoledronate (Cet-ZA) [30]. This novel ADC can trigger the expansion and activation of Vδ2 T cells targeting CRC primary organoids [30]. This effect was also detected when tumor-associated fibroblasts (TAF) from CRC mucosa were primed with Cet-ZA [31]. These findings suggest that Cet-ZA can be a useful tool to target N-BPs to tumor cells, limiting the preferential localization of soluble N-BPs into the bone [32]. Herein, we created two novel ADCs called cetuximab-ibandronate (Cet-IBA) and cetuximab-risedronate (Cet-RIS). This is to compare the effects of the different ADCs composed of cetuximab and N-BPs. These antibodies have been tested in several assays to determine their reactivity and functional effects on the proliferation of CRC cell lines as well as on the triggering of Vδ2 antitumor activity. This has been performed both in conventional and 3D culture assays to assess the efficiency of the different N-BPs and to plan therapeutic use in CRC.

## 2. Materials and Methods

### 2.1. Cell Lines and Cell Cultures

The human CRC cell lines Caco-2, Colo-320DM, Colo-205, DLD-1, HCT-116, LS-180, RKO, SW-48, and SW-620 were obtained from the Cell Bank of the Policlinico San Martino hospital while NCI-H716 was from ATCC (American Tissue Culture Collection). These CRC cell lines were authenticated by STR analysis and cultured in adherent plates with RPMI-1640 (Gibco, Life Technologies Italy, Monza, Italy) medium supplemented with 10% fetal serum (FBS, Gibco™ One Shot™ Fetal Bovine Serum, Thermo Fisher Scientific Italy, Monza, Italy), penicillin/streptomycin, and l-glutamine (BioWhittaker^®^ Reagents, Lonza, Basel, Switzerland) in a humidified incubator at 37 °C with 5% CO_2_. CRC cell line spheroids were generated as described [31] by culturing 2 × 10^4^ cells per well in flat-bottomed 96-well plates (Ultra-Low attachment multiwell plates, Corning^®^Costar^®^, New York, NY, USA) with DMEM-F12 (BioWhittaker^®^Reagents, Lonza) in serum-free medium (SFM), supplemented with EGF (Peprotech Europe, London, UK) at a 10 ng/mL final concentration (≥1 × 10^6^ IU/mg). Cell spheroid cultures were checked over time with the inverted microscope IX51 (Olympus, Hamburg, Germany) and were used for co-culture experiments with lymphocytes within 5–7 days as described in 2.7. Cell spheroid size was also monitored by time-lapse imaging, by a JuLi-Stage video history recorder (Nanoentek, Seoul, Republic of Korea). The vitality of cells was detected by the cell death probe C. Live Tox green (Cytena GmbH, Freiburg, Germany) on the whole spheroid, and quantified by image analysis. Parallel samples were analyzed after disaggregation and labeling with propidium iodide (PI, ThermoFisher, Milan, Italy).

### 2.2. Cet-IBA or Cet-RIS or Cet-ZA ADC Synthesis

IBA, RIS, and ZA were purchased from Selleckchem (Houston, TX, USA), and Cet was obtained as a leftover of the therapeutic preparation used for patients suffering from CRC (the kind gift from the Pharmacy Unit of IRCCS Ospedale Policlinico San Martino, Genoa, Italy). Cet-IBA, Cet-RIS, and Cet-ZA ADCs were prepared by Nanovex Biotechnologies (Asturias, Spain), according to the procedure reported before for the Cet-ZA (Figure 1) [30]. The protein sample was purified via ultrafiltration (100 kDa cut off) to eliminate the presence of excipients. An aliquot of a Cet sample was analyzed by ICP-MS to determine the sulfur (S) and the phosphorus (P) content. The purified fraction of the native antibody was employed in the reaction of conjugation with the corresponding bisphosphonate. The conjugation of the bisphosphonate to the protein was performed in imidazole 0.1 M solution pH 6 (adjusted with HCl). The bisphosphonate compound was incorporated into the structure of the protein via a reaction with 1-Ethyl-3-3-dimethylaminopropyl) carbodiimide (EDC), in the presence of imidazole. Appropriate amounts of Cet, the corresponding bisphosphonate, and EDC were mixed in imidazole media, and then the samples were allowed to react overnight. The excess of reagents of conjugation was eliminated by ultrafiltration (10 kDa cut off). An aliquot of each bisphosphonate-conjugated sample was analyzed by ICP-MS to determine the phosphorus content and, therefore, the bisphosphonate concentration. The reactions between the bisphosphates and Cet were performed with the molar excess of 759 bisphosphonates per protein for IBA or RIS and 712 for ZA. No linker molecule was involved in the covalent bond between Cet and either IBA or RIS or ZA.

### 2.3. Bioanalytical Characterization of ADCs

Matrix-Assisted Laser Desorption Ionization Mass Spectrometry (MALDI-MS) was performed to estimate the drug–antibody ratio (DAR) by the Fondazione Toscana Life Sciences in Siena, Italy. The desalting of the native Cet and the corresponding ADCs (Cet-IBA, Cet-RIS, or Cet-ZA) was carried out using SpinTrap G25 (Cytiva, Vancouver, BC, Canada). To prepare the samples for MALDI-MS analysis, amounts of 2 µL of desalted Cet or ADCs were mixed with 2 µL of a saturated solution of s-DHB in 0.1% TFA in acetonitrile:water (50:50, *v*/*v*). The mixture was then spotted on the MALDI target and left to dry in the air. Subsequently, 1 µL of the saturated matrix solution was added to each spot and left to dry. The mass spectra were acquired using a UltrafleXtreme (Bruker Daltonics, GmbH, Billerica, MA, USA) in linear mode, over the mass range m/z 30–220 kDa. Calibration was performed using a Protein Standard II Calibration Mixture (Bruker Daltonics, GmBH). The phosphorus or sulfur content of different samples was analyzed by ICP-MS (Inductively Coupled Plasma-Mass Spectrometry) technique by Nanovex, using an 8800 ICP-QQQ (Agilent Technologies, Milan, Italy). Samples were digested in acidic media (HNO_3_ 2%) prior to analysis. Analysis of phosphorus (P) content provided information about the bisphosphonate concentration, whereas the analysis of the sulfur (S) was related to the protein concentration in the sample.

### 2.4. Confocal Microscopy Analysis of the Localization of the Native Cet and Cet-ADCs in CRC Cell Lines

The Caco-2, DLD-1, or HCT-116 CRC lines (10^4^ cells/200 µL) were cultured for 24 h in clear flat-bottomed black-wall imaging 96-well plates (Eppendorf), in complete RPMI-1640 medium, at 37 °C in 5% CO_2_ atmosphere. Afterwards, 2 µg/mL of either native Cet or Cet-IBA or Cet-RIS or Cet-ZA ADCs were added, allowing the internalization and processing of the Cet-EGFR complex into the target cell for 24 h. The samples were then fixed in 70% ethanol/PBS for 20 min at 4 °C and incubated with either the anti-LAMP-1 mAb (clone H4A3, Thermo Fisher Scientific) or the anti-EEA1 mAb (clone 14/EEA1, Becton Dickinson Italia S.p.a., Milan, Italy) followed by the Alexa Fluor 647-α-hIg antiserum, the goat anti-mouse (GAM) anti-isotype specific Alexa Fluor 488 antiserum and Sytox Orange (50 nM, ThermoFisher). Samples were observed with the PlanApo 20X NA 0.80 air objective or with the PlanApo 40X NA 1.40 oil objective with the FV500 confocal Laser Scanning Microscope System (Olympus Europe GmbH, Hamburg, Germany). Each image was taken by sequential scanning and switching on the appropriate laser to allow the excitation of one fluorochrome at a time and to avoid the cross-contribution of the fluorochrome emission. The images were analyzed with FluoView 4.3b computer software (Olympus, Milan, Italy). Results are shown in pseudo-color as red (Alexa Fluor 647) or green (Alexa Fluor 488) fluorescence vs. nuclei in blue (Sytox Orange).

### 2.5. Proliferation Assay of CRC Cell Lines

Proliferation of the CRC cell lines was assessed by culturing 10^4^ cells of Caco-2 or DLD-1 or HCT-116 for the indicated period of time, in adherent 96-well flat-bottomed plates (Sarstedt, Germany). The amounts of Cet, Cet-IBA, Cet-RIS, or Cet-ZA ADC (2 µg/mL) were added during cell plating and the confluence of each well was evaluated by recording the cultures with a CellCyte X^TM^ scanner (Cytena GmbH, Germany) for 120 h. Images were taken every 12 h and the percentage of confluence and/or the area of the culture were quantified by the integrated software. CRC cells cultured without antibodies represented the control (CTR). Similar experiments were performed on spheroids, obtained by culturing 2.5 × 10^5^ cells/well in an AggreWell^TM^400 (StemCell^TM^, Vancouver, BC, Canada) 24-well plate for 48 h, in complete medium. Afterwards, the amounts of Cet, Cet-IBA, Cet-RIS, or Cet-ZA ADC were added and incubated for 120 h. Spheroid size was analyzed by the CellCyte X^TM^ scanner. This second method for spheroid generation showed the advantage of producing a large number of spheroids of similar size in a very short period of time.

### 2.6. Western Blot on CRC Cell Lines

To detect the effect of the native antibody Cet or the ADC on EGFR-mediated signaling, the Caco-2 and the HCT-116 cell lines were incubated in 5% CO_2_ at 37 °C in six-well plates for 24 h without or with 2 µg/mL of each antibody in complete medium. This experimental condition resembles the culture conditions applied in all the other experiments. Then, these incubation cells were lysed and subjected to Western blot (WB) analysis. WB was performed as described in detail [30]. Briefly, 30 µg/lane of total protein from 4 × 10^6^ CRC cell lines Caco-2 or HCT-116 extracted with RIPA buffer were run on 8% PAGE gels (Express Plus, GenScript, Piscataway, NJ, USA) and blotted on PVDF membranes (GE-healthcare, Milan, Italy). The following rabbit antibodies from Cell Signaling Technology (Danvers, MA, USA) were used: anti-p-EGFR 1:1000 (Tyr1068 clone D7A5), anti-EGFR 1:1000 (clone D38B1), anti-p-Akt 1:1000 (Ser473 clone D9E), anti-Akt 1:1000, anti-p-Erk1-2 1:2000 (Thr202/Tyr204), anti-Erk1-2 1:2000. The goat anti-rabbit HRP-conjugated secondary antibody 1:10,000 was used to detect antibody reactivity with the target protein and rabbit anti-β-actin HRP-conjugated 1:10000 was used as loading control. The chemiluminescent HRP substrate (Immobilon Western, Millipore S.p.a., Milan, Italy) was acquired using a Mini-HD9 Gel Documentation Systems (UVITEC, Cambridge, UK). Protein gel bands were quantitatively evaluated with the Image Studio Lite (v5.5.4) analysis software (LI-COR Biosciences GmbH, Hamburg, Germany), and phosphorylated proteins were normalized against their specific total protein content and β-actin.

### 2.7. Ex Vivo Expansion of Vδ2 T Cells

Buffy coat from healthy adult donors was collected and stratified on density gradient centrifugation using lymphocyte separation medium (Pancoll human, Density: 1.077 g/mL, PAN-Biotech, Munich, Germany), and peripheral blood mononuclear cells (PBMC) were obtained as described [16]. T cells were purified from PBMC by the RosetteSep negative selection kit for T cells (Stem Cell Technologies, Vancouver, BC, Canada) as described [30]. The resulting population of leukocytes was composed of >95% T cells and the absence of residual CD14+ monocytes was verified by specific immunofluorescence [30]. The assessment of Vδ2 T cell expansion by the ADC in co-cultures of purified T cells and the CRC cell line LS-180 were performed as described in detail [33]. LS-180 is a cell line that can induce a strong expansion of Vδ2 T cells when incubated with soluble N-BPs such as ZA. Briefly, 10^6^ cells/mL purified T cells were co-cultured with 10^4^ LS-180 CRC cells in 200 µL of complete RPMI-1640 medium in 96-well U-bottom plates, with or without Cet or ADC 2 µg/mL, for 24 h. After the incubation, and every 3 days, 100 µL of culture medium was discarded and 100 µL of human- and animal-free recombinant interleukin 2 solution (IL-2, Cat. No. AF-200-02 Peprotech) was added to each well (30 IU/mL final concentration). The Vδ2 expression was determined at different time points (10 and 20 days of culture) by indirect immunofluorescence assay using the anti-Vδ2 antibody, clone γδ123R3, IgG1 isotype (Miltenyi Biotech, cat. no. 130-121-139, or home-made supernatant). Samples were run on a CytoflexS flow cytometer (Beckman-Coulter, Brea, CA, USA) and results analyzed by the CytExpert 2.4 software. In an additional series of experiments, we used cell suspensions derived from CRC patients (OMCR12-040, OMCR12-041, OMCR12-052, OMCR12-065, OMCR18-009) after the digestion of tumor specimens as described [30]. After the thawing of samples, cell suspensions were analyzed for the presence of Vδ2 T cells. A total of 10^5^ cells were then cultured with or without Cet or ADC at 2 µg/mL as above and IL-2 added after 24 h. Cell cultures were followed and split in IL-2 when cell growth was microscopically evident. The percentage of Vδ2 T cells was analyzed at different time points. A strong heterogeneity in cell growth was found, but usually all the cultures could be assessed for the presence of Vδ2 T cells by immunofluorescence at day 25 and compared to the initial percentage of Vδ2T cells. Biological specimens from healthy donors were obtained after signed informed consent by the blood transfusion center at the moment of donation while those for CRC patients were obtained after signed informed consent according to the Ligurian Regional Ethic Committee approval, PR163REG2014, renewed in 2017.

### 2.8. Cytotoxicity Assay

To evaluate the in vitro cytotoxicity of ADCs, we performed experiments on the DLD-1, HCT-116, and Caco-2 CRC cell lines either in conventional culture conditions or as spheroids. Vδ2 T lymphocytes were obtained from PBMC as described [26], and their expressions of Vδ2 TCR and CD16 were analyzed using specific mAbs and flow cytometry analysis (90–95% Vδ2+, 50–75% CD16+). The Vδ2 effector cells were incubated with either adherent CRC cells or spheroids at an E:T ratio of 2:1 for 48 h. Cet (2 µg/mL) or ADCs (2 µg/mL) were added to the cultures from the beginning of the assay. The addition of Cet or ADCs allowed us to evaluate the contribution of the bisphosphonate drug to the cytotoxicity of the Vδ2 effector cells. To evaluate cytotoxicity, the crystal violet assay was performed as previously described [26]. This assay quantifies the amount of surviving CRC cells at the end of the assay. Eluted crystal violet from labeled cells was measured using the VICTORX5 multilabel plate reader (Perkin Elmer, Milan, Italy), at 594 nm. The results are shown as OD, or as a percentage of the OD of the control wells, represented by the spheroids cultured without the addition of Vδ2 effector cells, as previously described [26].

### 2.9. Statistical Analysis

Data are presented as mean ± SD. Statistical analysis was performed using two-tailed unpaired Student’s *t*-test, using the GraphPad Prism software 9.4.0. The cutoff value of significance is indicated in each figure legend.

## 3. Results

### 3.1. Synthesis of Cet-IBA, Cet-RIS, and Cet-ZA ADCs

The drugs RIS, IBA, or ZA were covalently linked to Cet according to the previously described protocol [30], based on the reactions among the phosphoric groups of nucleic acid and the free amino groups of the antibodies. The MALDI analysis of Cet and ADCs (Cet-IBA, Cet-RIS, and Cet-ZA) confirmed the creation of covalent bonds in the ADCs, as evidenced by the increase in molecular weight of drug-loaded antibodies compared to native mAb (Figure 2). To quantify the amount of bisphosphonates per monoclonal antibody, the drug–antibody ratio (DAR) was computed for all ADCs by dividing the difference between the masses of the conjugated and unconjugated antibodies by the anticipated mass change upon the conjugation of one mole of the corresponding drug (IBA or RIS or ZA). The average DAR corresponded to 5.7 or 7.1 or 3.5 for Cet-IBA, Cet-RIS, or Cet-ZA, respectively. To further confirm the presence of N-BPs linked to the antibody, the sulfur (S) and phosphorus (P) contents of the ADCs were analyzed by ICP-MS, to determine the P/S molar relation in the conjugated proteins. The S content refers to Cet, while the P content refers to N-BPs. Considering the molecular weight of each bisphosphonate (BP) and protein and the amount of P and S per molecule, the P/S molar ratios in the conjugated proteins are reported in the Table 1 and Figure 2. Table 1 reports the molarity of the N-BPs in the solution of the ADC, used in phenotypic and functional experiments. Overall, these findings indicate that the N-BPs are covalently linked to the anti-EGFR antibody.

### 3.2. Characterization of ADCs

To investigate the reactivity of the ADCs towards EGFR-expressing CRC cell lines, such as DLD-1 and HCT-116, we incubated the cells with serially diluted ADCs, followed by APC-labeled anti-human Ig antiserum and flow cytometry analysis. The results indicate that Cet-IBA, Cet-RIS, and Cet-ZA had a reactivity comparable to the native Cet. The optimal reactivity dose of 2 µg/mL/10^6^ cells was selected for functional experiments. Some flow cytometry histograms are shown in Figure 3B. It is of note that the pattern of reactivity of the ADC was superimposable onto that of the native Cet. Indeed, the fluorescence intensity was similar both in low-expressing (SW-620) and high-expressing (NCI-H716) EGFR^+^ cell lines (Figure 3B).

### 3.3. ADC Internalization in CRC Cell Lines

We assessed whether the aminobisphosphonate-conjugated antibodies could be internalized. Indeed, a requisite for the functional activity of such an ADC is the entry into the cell upon interaction with EGFR. CRC cell lines were incubated with one ADC for 24 h. After fixation, cells were labeled with anti-LAMP-1 or EEA-1 antibodies to identify the co-localization of EGFR with lysosomes or early endosomes. Sytox Orange probe was used to label the nuclei, and cells were analyzed by confocal microscopy. As shown in Figure 4, the cell region colored as yellow is the overlapping of green (EEA-1 or LAMP-1) and red fluorescence (native Cet and each ADC) suggesting co-localization among lysosomes or endosomes and the internalized EGFR.

These co-localizations were more evident for the HCT-116 cell line, compared to DLD-1 or Caco-2 (Appendix A). These differences may be attributed to a different EGFR turnover in the three CRC cell lines analyzed. These findings indicate that the Cet-IBA, Cet-RIS, and Cet-ZA ADCs, after interaction with the EGFR, can be internalized and processed in endosomal and lysosomal compartments.

### 3.4. ADC Effect on CRC Cell Line Proliferation

The ADCs were tested for a possible direct effect on the proliferation of the Caco-2, DLD-1, and HCT-116 cell lines, using two different experimental settings. These cell lines were selected as Caco-2 shows a wild-type EGFR-mediated signaling; thus, it is expected to be a Cet-responder, while DLD-1 and HCT-116 are affected by KRAS mutations acting downstream of EGFR activation [34]. In the first set of experiments, cells were seeded in conventional adherent plates and incubated with the ADC, or the native antibody Cet. The cell cultures were monitored with a cell culture scanner for 120 h, recording a picture every 24 h. We found that Cet can halve Caco-2 cell proliferation (Figure 5A). Interestingly, the Cet-N-BP ADCs can amplify this inhibition, reaching about 90% after 120 h (Figure 5A). On the other hand, neither the native antibody nor the ADC can inhibit the proliferation of HCT-116 (Figure 5A) or DLD-1 (Appendix A), Colo-205, Colo-320DM, and SW-48. This finding would indicate that IBA, RIS, and ZA can influence cell proliferation synergizing with the anti-EGFR therapeutic antibody. Notably, aminobisphosphonates in soluble form can affect the proliferation of all these cell lines, although at a different rate and at higher doses (see Appendix A). Indeed, ZA 25–50 µM inhibited Caco-2 growth by 50% (Appendix A), while the amount of ZA (and the other N-BPs) linked to Cet was present at nanomolar concentrations (see Table 1). No free N-BP affected these cell lines at 1 µM concentration. These findings would suggest that the direct efficacy of the N-BPs is markedly increased if delivered through the antibody. To investigate whether the ADC can also affect the proliferation of Caco-2 when grown as small tumor masses, we cultured this cell line in AggreWell^TM^400 plates without (CTR) or with the native Cet or ADC. Under these culture conditions, several tumor spheroids with similar sizes can be generated. This culture system allows the generation of several individual spheroids (about 1200 for each well in a 24-well plate), whose size can be monitored at different time points by image recording. The results obtained after 5 days of culture are shown in Figure 5B. Overall, the inhibitory effect of native Cet was about 15%, while the inhibition triggered by the RIS- or ZA-linked ADC exceeded 50%. On the other hand, Cet-IBA was less effective on Caco-2 spheroid size (about 20%). The reduction in Caco-2 spheroids’ growth was also confirmed by evaluating their ATP content. These findings indicate that N-BPs linked to Cet can increase the inhibitory effect of the native antibody on cell proliferation.

### 3.5. Cet-ZA, Cet-IBA, and Cet-RIS ADCs Can Trigger Vδ2 T-Cell-Mediated Antitumor Cytotoxicity

It is well known that N-BPs can trigger Vδ2 T cell antitumor activity upon presentation, by tumor cells, of small antigen pyrophosphates [30,31,32,33]. We have already demonstrated that Cet-ZA can elicit the antitumor cytolysis of organoids derived from primary CRC specimens. Thus, we needed to confirm if the novel Cet-IBA and Cet-RIS ADCs could elicit a comparable response of Vδ2 T cells against Caco-2, DLD-1, and HCT-116. We used the 3D culture system composed of spheroids to mimic a small tumor mass. Spheroids of different sizes were obtained in ultra-low adherent plates, as previously described [26]. In preliminary experiments, the IC50 of each antibody was determined on HCT-116 and Caco-2, in the cytotoxicity assay, to assess the potency of the ADCs (Appendix A). Cet-ZA ADC was found to be the most effective, followed by Cet-RIS and Cet-IBA, respectively. Despite having the lowest DAR (3.6), Cet-ZA showed the lower IC_50_ for both cell lines, as shown in the Appendix A.

Based on phenotypic and IC_50_ values, a single effective dose of ADCs (2 µg/mL) was used for cytotoxic experiments, with all cell lines. Vδ2 T cells generated from the PBMC of three distinct healthy donors were added to CRC spheroid cultures at a 3:1 E:T ratio, along with ADCs. After a 48 h incubation, the spheroids were transferred into conventional flat-bottomed plates to allow cell adherence. After an additional 24 h, crystal violet staining was performed to quantify the surviving CRC cells (Figure 6). 

Among the ADCs tested, Cet-ZA demonstrated the highest cytotoxicity against CRC spheroids. Cet-RIS and Cet-IBA were, respectively, the second and third most potent ADCs in enhancing cytotoxicity. Notably, the cytotoxicity observed against Caco-2, the cell line with a wild-type EGFR-pathway, was less efficient as compared to the killing of the DLD-1 and HCT-116 cell lines. This effect was also evident in the absence of Cet or ADC. Some heterogeneous responses were observed among the different donors of Vδ2 T cells and the ADCs did not always elicit a statistically significant increase in the basal Vδ2 T cells’ cytotoxicity.

### 3.6. ADC Can Trigger Vδ2 T Cell Expansion

The impact of ADCs on the expansion of Vδ2 T cells isolated from PBMC was evaluated. In fact, soluble N-BPs can induce the activation and subsequent proliferation of Vδ2 T cells in the presence of exogeneous IL2. To determine whether N-BPs linked to Cet can yet trigger Vδ2 T cells’ proliferation, co-culture experiments with the CRC cell line LS-180 were performed. In fact, we previously showed that not all the CRC cell lines can stimulate Vδ2 T cells’ activation, while LS-180 represents an optimal activator, able to present small pyrophosphate antigens leading to IL2-supported Vδ2 T cell expansion [33]. Highly purified T cells from healthy donors (>95% CD3^+^, <0.5% CD14^+^) were co-cultured with irradiated LS-180 cells at the 20 (T):1 (LS-180) ratio, with 2 µg/mL of single ADC or native Cet. Cultures without antibody were used as control. After 10 and 20 days, cell cultures were assessed for the presence of Vδ2 T cells by immunofluorescence assay. As shown in Figure 7, a marked increment in the percentages of Vδ2 T cells was detected starting from day 10; on day 20 this population was predominant among T-LS-180 co-cultures with the ADC (Figure 7A,B). It is of note that, also in this assay, the Cet-IBA ADC was less efficient compared to Cet-RIS and Cet-ZA.

We further analyzed whether the ADC could trigger the expansion of Vδ2 T cells derived from CRC tumors. Frozen cell suspensions from primary CRC specimens, obtained by enzymatic digestion, were thawed and put in culture. The analysis for Vδ2 T cells found in fresh cell suspensions from CRC patients revealed that this population was barely detectable (0.01–0.5% of the whole cell suspension). Importantly, a detectable increment in Vδ2 T cells was found when these cell suspensions were cultured with the ADC and IL-2, reaching the maximal expansion after 25 days (Figure 7C). Again, Cet-RIS and Cet-ZA triggered a stronger expansion than Cet-IBA. Altogether, these data would suggest that N-BPs carried by anti-EGFR antibodies were processed by CRC cells and efficiently stimulated Vδ2 T cell growth.

## 4. Discussion

In this study, we have shown the successful development of new ADCs. These ADCs are created by combining N-BPs (either IBA or RIS) with the humanized anti-EGFR mAb Cet. This development was based on our previous report on Cet-ZA ADC [30].

Herein, we have shown that the N-BPs linked to Cet, likewise soluble N-BPs, can stimulate anti-CRC effects [26]. Also, this formulation of N-BPs can increase the inhibitory effect of Cet on the proliferation of the EGFR WT CRC cell line Caco-2. Furthermore, these ADCs can stimulate the expansion of Vδ2 T cells, when processed by the CRC cell line LS-180. Altogether, these findings suggest that these ADCs can exert a stronger antitumor effect by two mechanisms: (1) directly affecting tumor cell proliferation and (2) indirectly activating an immune response. In this regard, our study has a limitation as the inhibition of cell growth was clearly evident only with Caco-2 cells and not with HCT-116 or DLD-1. Actually, these two cell lines may bypass the EGFR-mediated signaling through KRAS mutations, able to promote cell proliferation without EGFR triggering, as reported [34]. The direct regulation of cell proliferation by our ADCs should be confirmed on a larger panel of WT-EGFR pathway CRC cell lines.

It should be noted that the cell lines used in this work show the activating mutation of signaling molecules that render them mostly independent of the presence of growth factors and EGFR-mediated signal transduction. Indeed, the presence of compensatory signals, substituting the EGFR, is present in each of the CRC cell lines analyzed (http://cancerres.aacrjournals.org/content/suppl/2014/04/22/0008-5472.CAN-14-0013.DC1/Tab3.xls, accessed on 13 February 2024). According to this consideration, we did not find an evident effect of either native Cet nor ADC on the amount and phosphorylation of EGFR and downstream signaling molecules (Appendix A). Overall, we cannot state that the antibody treatment led to evident effects on the total EGFR, Akt, and Erk1 and Erk2 as well as their phosphorylated forms. This could have been expected in the absence of an added ligand. Actually, it appeared that pAkt is slightly increased in the Caco-2 cell line when treated with Cet-ZA compared to the other antibodies (Appendix A). This can be a compensation mechanism to favor Caco-2 cell survival. Also, this effect was not evident in the HCT-116 cell line. The densitometric analysis did not reveal clear differences (more or less than two-fold of basal levels). These finding would suggest that under our experimental conditions (absence of EGF, not starved cells, culture with medium with FCS) the EGFR signaling is not clearly induced or altered (indeed, the EGFR p1068 phosphorylation signal was barely detectable, Appendix A).

This experimental context, would suggest that the EGFR can be considered as a target molecule expressed at the cell surface of CRC cells, independently from its effective ligand-triggering activity. Indeed, all our tests were performed without the addition of any EGFR ligand. When engaged by the ADC linked to the N-BP, this receptor can be partly endocytosed and can deliver the N-BP into the cell. Afterwards, the N-BP acting on the mevalonate pathway leads to IPP production and the activation of Vδ2 T cells. The dissociation between the antibody and the N-BP should take place in the lysosomes due to the acidic milieu. Eventually, the N-BP leads to pAg production and its presentation to Vδ2 T cells. Actually, we cannot exclude that EGFR engagement with the native or N-BP-linked antibody can deliver a signal inside the CRC cells. But all the CRC cell lines tested in our study can proliferate independently from EGFR, because of the above-mentioned compensatory mutations. The inhibitory effect of anti-EGFR on Caco-2 cell proliferation would suggest that this cell line can still depend on EGFR-mediated signaling. This could be explained with the observation that Caco-2, producing the EGFR ligand such as epiregulin (EREG) [35], can trigger EGFR-mediated autocrine proliferation. If this is the case, the antibody could affect the interaction between EREG and EGFR, reducing the proliferative effect. The stronger effect of N-BP-linked antibodies could be explained by the apparently stronger reactivity with the EGFR shown by these ADCs compared to the native antibody at least with some CRC cell lines.

Also, it is known that the interference of EGFR-mediated signaling by antibodies can affect the activation of other signaling molecules leading to resistance to therapy and to error-prone DNA replication [35,36,37,38,39,40]. In our experiments, we did not detect a clear downregulation of the EGFR expression (Appendix A) but we can observe the endocytosis of the anti-EGFR antibodies (Figure 4). This would suggest that other activation pathways are not triggered, but also, we cannot exclude that the ADC would favor the generation of resistant tumor cells upon treatment. If this should be the case, receptors such as HER2, HER3, IGF1R, and AXL [35,36,37,38,39] could be considered as suitable targets for specific antibodies conjugated to N-BPs.

N-BPs can trigger the activation of Vδ2 T cells via the specific TCR engagement. Indeed, the small pyrophosphate antigens produced in either antigen-presenting cells or tumor target cells can trigger either the proliferation of Vδ2 T cells or the killing of tumor target cells [24,33,40,41,42,43,44,45]. We have found that cetuximab linked to either RIS or ZA can trigger a stronger direct or indirect effect on CRC cells than Cet-IBA. Actually, Ibandronate is a less potent N-BP than RIS or ZA, suggesting that the different degree of functional effects detected could be related to the ability of a given N-BP in the inhibition of FFPP enzymes and the generation of small pyrophosphate antigens [28,45]. Whatever the reason of the different potency of the ADC tested, it is evident that the net amount of N-BPs that can enter a cell when linked to the ADC is in nM concentrations, while soluble ZA can show some effects on tumor cells at µM concentrations. This finding is similar to what we reported previously using nanovectors to target tumor cells [46]. Taken together, these observations indicate that the use of either ADCs or nanoparticles charged with N-BPs could improve the subcellular concentration and the consequent metabolism of the N-BPs, leading to a stronger functional effect than its soluble form.

It is well known that a major drawback of the administration of N-BPs such as ZA, for pathologies not involving bones, is their tendency to rapidly accumulate in the bones, disappearing from circulation [31]. As a result, their concentration in the bloodstream is very low, as the unbound quota is excreted in the urine. ADCs are a rapidly growing approach in cancer treatment and are often referred to as “biological missiles” for cancer therapy [11,35]. This treatment involves the linking of a monoclonal antibody (Ab) to target cancer cells with a cytotoxic payload [5,6,7,8].

EGFR is a tyrosine kinase that belongs to the Erb B family and plays a role in regulating cell proliferation and apoptosis resistance through multiple signal transduction pathways [47,48,49]. Overexpression of EGFR has been observed in several types of cancer [47,48,49]. In this study, we found that ADCs can specifically deliver aminobisphosphonate to cancer cells that consistently express EGFR. This delivery modifies the distribution of the drug and ultimately triggers the body’s anti-cancer response by activating Vδ2 T cells.

While Cet is used in the clinic against mCRC only in the absence of EGFR pathway mutations, here we show that the novel ADCs are functional in both KRAS- and PIK3CA-mutated DLD1 and HCT116 cell lines (http://cancerres.aacrjournals.org/content/suppl/2014/04/22/0008-5472.CAN-14-0013.DC1/Tab3.xls, accessed on 13 February 2024). These alterations make these cell lines independent of cell growth regulation by EGFR-mediated signaling. Therefore, these findings suggest that the activation of Vδ2 T cells through ADCs can be effective in any CRC cell line expressing EGFR, regardless of the presence of specific mutations. During the analysis of cytotoxicity induction by ADCs, we observed that Cet-ZA and Cet RIS were particularly strong in triggering cytotoxic effects. This is attributed to the engagement of CD16 and TCR. Cet-ZA and Cet-RIS exhibited an increased cytotoxicity co pared to Cet-IBA. This difference in potency was due to their higher efficiency in inhibiting FPPS, which led to the accumulation of IPP and the activation of Vδ2 T cells. Based on previous studies, the potency in inhibiting FPPS was higher in ZA, followed by RIS and IBA, respectively [rate affinity]. As a result, the use of these ADCs to activate Vδ2 cells that infiltrate the CRC mucosa could be effective in a wide range of cases. These findings provide valuable insights into the efficacy of ADCs in targeting EGFR-e pressing CRC cells and highlight the contribution of the bisphosphonate drug to their potency. The results obtained from our study can be utilized to optimize the design of ADCs and guide their further development for clinical applications. However, this could be planned after the demonstration of the efficacy of these novel ADCs together with the lack of toxicity for healthy tissues in appropriate murine models.

## 5. Conclusions

Different N-BPs covalently linked to the therapeutic anti-EGFR antibody cetuximab can further enhance the antitumor effect mediated by the native antibody. These ADCs increment the direct inhibitory effect on cell proliferation induced by the native antibody. Also, they can trigger both the proliferation of antitumor effector Vδ2 T cells and enhance the killing of CRC tumor cell lines mediated by Vδ2 T cells. Altogether, these findings suggest the possibility of planning the future use of these novel ADCs to target CRC cells in a clinical setting.

## Figures and Tables

**Figure 1 cancers-16-01256-f001:**
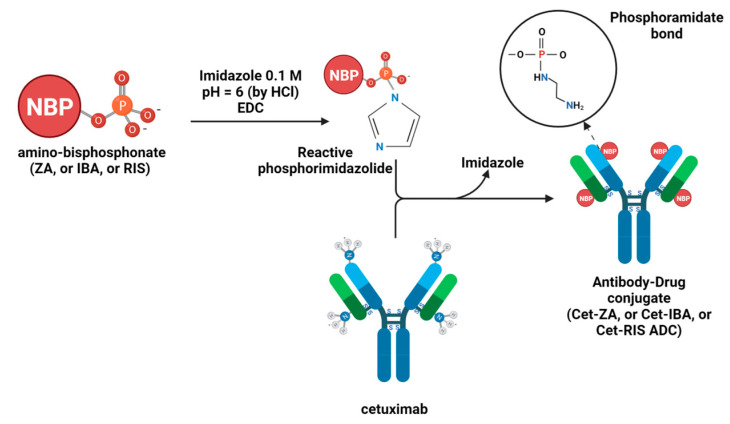
Schematic representation of the chemical reactions to synthesize Cet-IBA, Cet-RIS, or Cet-ZA ADCs [30].

**Figure 2 cancers-16-01256-f002:**
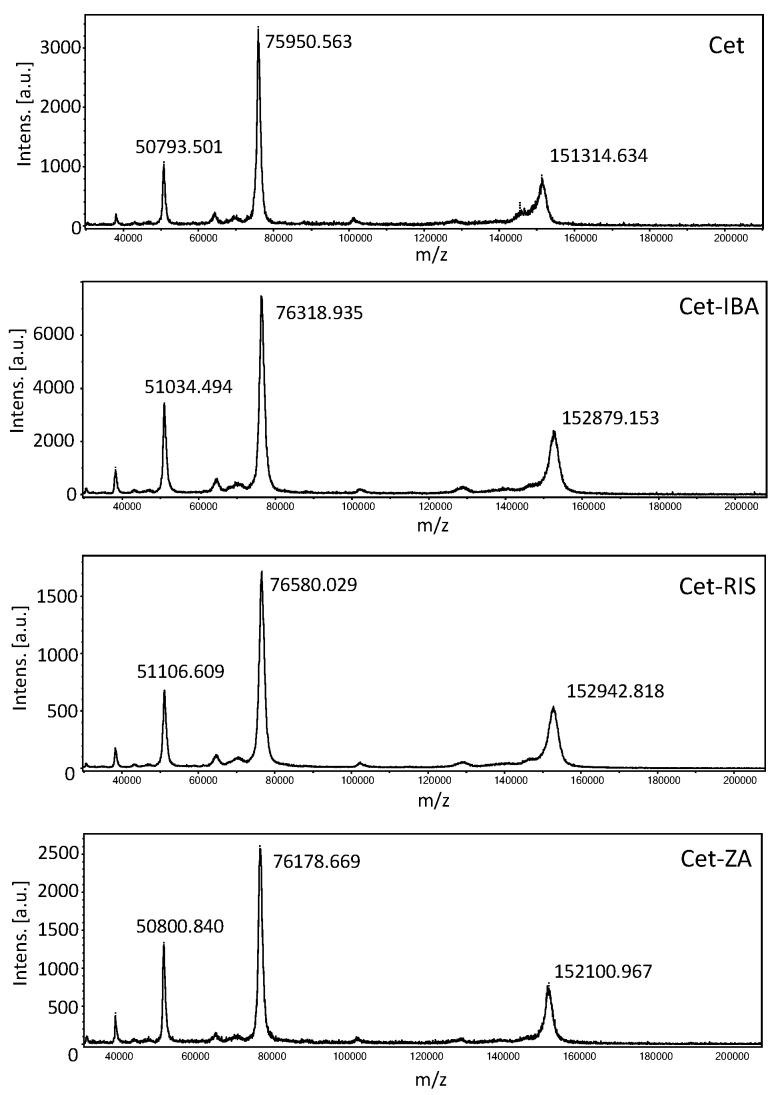
Matrix-assisted laser desorption ionization (MALDI) mass spectra of Cetuximab (**top**), Cet-IBA, Cet-RIS, and Cet-ZA ADCs (**bottom**) in order. In each panel, the molecular weight (MW) of singlet, doublet, and triplet ionized molecules are shown. The differences in MW are the evidence of the covalent conjugation of each N-BPs to the antibody Cet. The MALDI spectrum for native Cet is shown in the first panel.

**Figure 3 cancers-16-01256-f003:**
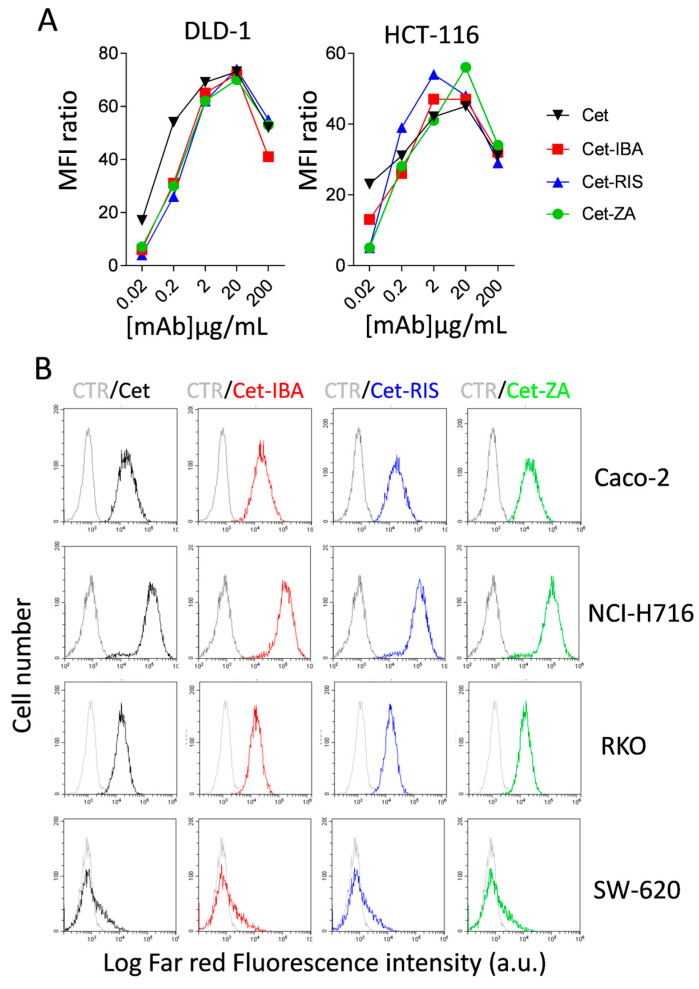
ADCs’ characterization. ADCs’ titration and reactivity with CRC cell lines. (**A**) The CRC cell lines DLD-1 and HCT-116 were incubated with serial dilutions (0.02–200 µg/mL/10^6^ cells) of Cet (black), Cet-IBA (red), Cet-RIS (blue), or Cet-ZA (green) followed by the APC-labeled anti-human Ig antiserum. Samples were analyzed by flow cytometry using CytExpert software 2.4. The data are expressed as the ratio of mean fluorescent intensity (ratio between the MFI of either Cet or the indicated ADC and the negative control). (**B**) FACS histograms: reactivity of either Cet (black) or the indicated ADC (2 µg/mL/10^6^; Cet-IBA red, Cet-RIS blue, Cet-ZA green) with the CRC cell lines Caco-2, NCI-H716, RKO, or SW-620. Data are shown as log far red fluorescence intensity vs. cell number. In each FACS histogram, in gray color is shown the negative control (CTR, cells stained with second reagent only).

**Figure 4 cancers-16-01256-f004:**
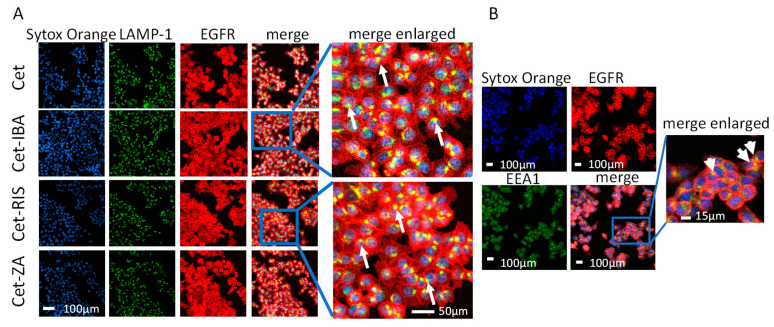
ADCs’ internalization and localization. (**A**) The HCT-116 cell line was seeded in imaging-specific 96-well flat-bottomed plates for 24 h to allow adherence. Then, the cells were incubated with the native Cet antibody (upper images) or ADC (lower images, Cet-IBA, Cet-RIS, or Cet-ZA) for an additional 24 h. Afterwards, cells were fixed and incubated with the anti-LAMP-1 antibody followed by staining with Alexa Fluor 488 (green color, second line of images), Alexa Fluor 647 (red color, anti-human to detect EGFR antibody, third line of images) secondary antibodies, and Sytox Orange probe to identify nuclei (blue color, first line of images). Merged images represent the overlay of the three stainings. Each image was taken at the confocal plane after scanning in sequence mode, to avoid cross-talk among the different wavelength emissions. The yellow region represents the co-localization area for the indicated markers. Bar: 100 µm, 200× magnification. (**B**) The HCT-116 cell line was seeded as in panel A and incubated with Cet-RIS for 24 h. Then cells were fixed and incubated with the anti-EEA-1 antibody. The yellow region, evidenced by the white arrows, indicates the overlapping area for EGFR and EEA-1. Bar: 100 µm, 200× magnification. Enlargement 10× of merge image.

**Figure 5 cancers-16-01256-f005:**
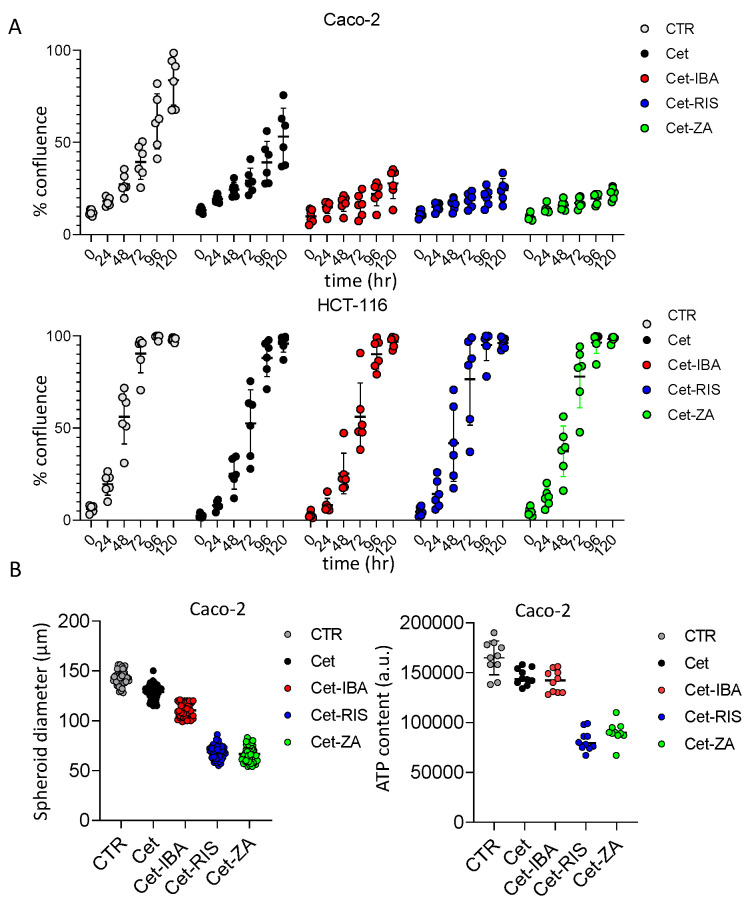
Effect of ADC on CRC cell line proliferation. (**A**) The indicated cell lines (Caco-2 and HCT-116) were seeded in flat-bottomed plates without (CTR, no antibody) or with 2 µg/mL of the indicated antibodies. Cell proliferation was assessed as the percentage of confluency by taking images of culture wells at the indicated time points (0, 24, 48, 72, 96, 120 h). Each point corresponds to six replicates for each experimental condition. The percentage of confluency was calculated by CellStudio 6.2 software, associated with the CellCyte X^TM^ plate scanner. The results are expressed as mean ± SD of confluency. (**B**) left panel: the spheroid diameter was measured in 50 spheroids obtained with AggreWell-400 plates after 5 days of culture without (CTR) or with 2 µg/mL of the indicated antibodies. (**B**) right panel: the ATP content of spheroids cultured as in left panel. Results are shown as ATP content expressed in arbitrary units.

**Figure 6 cancers-16-01256-f006:**
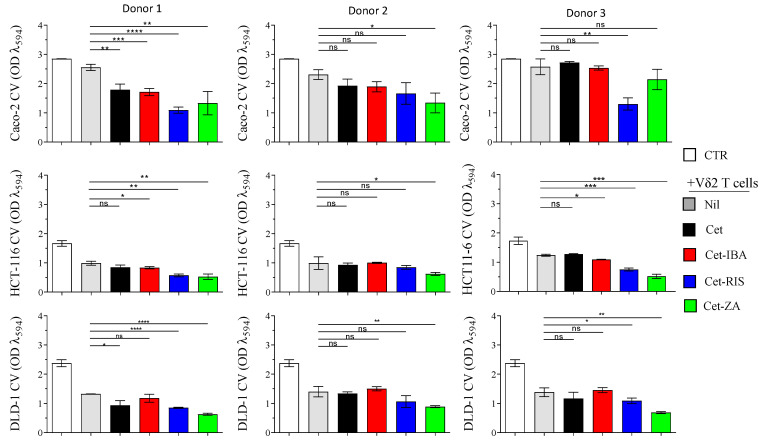
Triggering of Vδ2^+^T cell cytotoxicity with either ADCs or native Cet antibody. Spheroids of Caco-2, DLD-1, or HCT-116 were co-cultured with Vδ2 T cells, either in the presence of 2 µg/mL Cet, Cet-ZA, Cet-IBA, or Cet-RIS, or without treatment, at the effector:target ratio of 3:1. After 48 h, the cell cultures were transferred to adhesion-permissive plates, allowed to adhere for 24 h, and stained with crystal violet to quantify surviving/adherent cells. The optical density (OD) of the eluted stain was evaluated by a VICTORX5 multilabel plate reader at 594 nm. CTR: OD of cultured tumor cells only. Bars: mean ± SD of each group of measures. * *p* > 0.05; ** *p* > 0.01; *** *p* > 0.001 **** *p* > 0.0001, n.s. Not significant.

**Figure 7 cancers-16-01256-f007:**
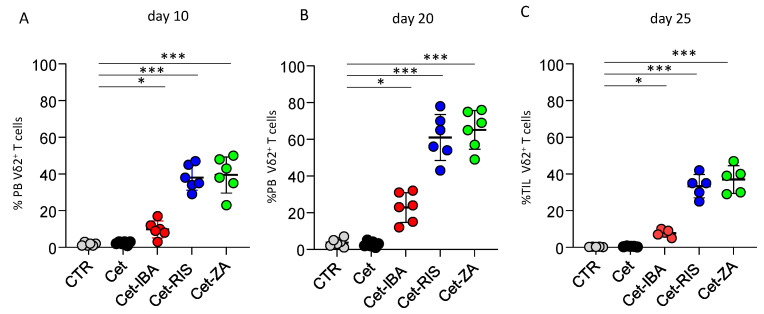
Induction of Vδ2 T cell expansion. Highly purified T cells, incubated with LS-180 CRC cells from the onset of assay with medium (CTR), or the indicated antibodies, were assessed for the presence of Vδ2^+^T cells by indirect immunofluorescence. Vδ2 T expression was determined at day 10 (**A**) and 20 (**B**) by indirect immunofluorescence flow cytometry, using the anti-Vδ2 antibody, clone γδ123R3, and plotted by GraphPad Prism software. (**C**) Cell suspensions from 5 CRC mucosa specimens were incubated as in (A) with the indicated antibodies and IL-2 was added after 24 h of culture. The Vδ2 T cell percentages were determined at the onset of the culture (CTR) and on day 25 of incubation with the different antibodies. Bars: mean ± SD of each group of measures. * *p* > 0.01; *** *p* > 0.001.

**Table 1 cancers-16-01256-t001:** Bioanalytical characterization of ADCs.

ADC	DAR	ppm P *	ppm S °	mol P/mol S	nM N-BPs/mL 2 µg/mL ADC **
Cet-IBA	5.7	65 ± 2	40 ± 4	1.7	74.1
Cet-RIS	7.1	74 ± 2	35 ± 1	2.2	92.3
Cet-ZA	3.6	16 ± 2	38 ± 1	0.4	46.8

DAR: drug–antibody ratio; ° part per million of either phosphorus (*) or sulfur (°). ** Concentration of the N-BPs linked to Cet in the solution used to assess antibody reactivity and functional effects.

## Data Availability

The data and reagents will be available under appropriate and motivated requests upon a material transfer agreement between the IRCCS Ospedale Policlinico San Martino and the other party.

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
