# Peer review of "EGFR-Targeted Antibody–Drug Conjugate to Different Aminobisphosphonates: Direct and Indirect Antitumor Effects on Colorectal Carcinoma Cells"

_cancers, 2024, doi:10.3390/cancers16071256_

Round 1

Reviewer 1 Report

Comments and Suggestions for Authors

The study explores the potential of antibody-drug conjugates (ADCs) by linking Cetuximab (Cet), a monoclonal antibody targeting EGFR, to aminobisphosphonates (N-BPs) such as ibandronate (IBA), risedronate (RIS), or zoledronate (ZA). These ADCs are evaluated for their impact on CRC cell line proliferation and sensitivity to Vδ2 T anti-tumor lymphocytes, both in vitro and in 3D spheroid cultures. Results indicate that all three ADCs enhance the inhibitory effect on cell proliferation in WT-EGFR cell lines, while Cet-RIS and Cet-ZA specifically enhance cytotoxicity mediated by Vδ2 T cells against both WT and EGFR-mutated CRC cell lines. Furthermore, the ADCs trigger proliferation of Vδ2 T cells in peripheral blood and tumor specimens. While the study represents an exciting advancement in cancer therapy, further experiments or discussions on the mechanisms underlying the ADCs' effects would provide deeper insights into their therapeutic potential.

Major Comments:

1. Inhibition of EGFR upregulates several receptors such as HER2 HER3 IGF1R and AXL as a compensatory mechanism. What is the status of these receptors when ADCs targeting EGFR are used? Authors have to discuss this in the manuscript.

2. Down regulation of EGFR using antibodies upregulates phospho-AXL levels. Authors have to monitor for pEGFR and pAXL levels using western blots.

3. Inhibition of EGFR upregulates low-fidelity polymerases. Authors to monitor the expression of low fidelity polymerases upon treatment with ADCs.

4. What is the status of EGFR when treated with ADCs? Authors have to check for RNA and protein expression of EGFR.

5. Authors have to discuss the intracellular effects of ADCs when EGFR is used as a target. Authors have to differentiate the molecular mechanism of cetuximab and ADCs.

Author Response

Reviewer 1

The study explores the potential of antibody-drug conjugates (ADCs) by linking Cetuximab (Cet), a monoclonal antibody targeting EGFR, to aminobisphosphonates (N-BPs) such as ibandronate (IBA), risedronate (RIS), or zoledronate (ZA). These ADCs are evaluated for their impact on CRC cell line proliferation and sensitivity to Vδ2 T antitumor lymphocytes, both in vitro and in 3D spheroid cultures. Results indicate that all three ADCs enhance the inhibitory effect on cell proliferation in WT-EGFR cell lines, while Cet-RIS and Cet-ZA specifically enhance cytotoxicity mediated by Vδ2 T cells against both WT and EGFR-mutated CRC cell lines. Furthermore, the ADCs trigger proliferation of Vδ2 T cells in peripheral blood and tumor specimens. While the study represents an exciting advancement in cancer therapy, further experiments or discussions on the mechanisms underlying the ADCs' effects would provide deeper insights into their therapeutic potential.

Major Comments:

  1. Inhibition of EGFR upregulates several receptors such as HER2 HER3 IGF1R and AXL as a compensatory mechanism. What is the status of these receptors when ADCs targeting EGFR are used? Authors have to discuss this in the manuscript.

We have considered this point in the M and M methods (line 156-204) and Discussion section (lines 482-507) as follows:

  1. It should be noted that the cell lines used in this work show activating mutation of signaling molecules that render them mostly independent on the presence of growth factors and EGFR-mediated signal transduction. Indeed, the presence of compensatory signals, substituting the EGFR, is present in each of the CRC cell lines analysed (http://cancerres.aacrjournals.org/content/suppl/2014/04/22/0008-5472.CAN-14-0013.DC1/Tab3.xls). According to this consideration, we did not find an evident effect of either native Cet or ADC on the amount and phosphorylation of EGFR and downstream signaling molecules (supplementary figure 5). This experimental context, would suggest that the EGFR can be considered as a target molecule expressed at the cell surface of CRC cells, independently of its effective ligand-triggering and activity. Indeed, all our tests were performed without the addition of any EGFR ligand. When engaged by the ADC linked to the N-BP, this receptor can be partly endocytosed and deliver into the cell the N-BP. Afterwards, the N-BP acting on the mevalonate pathway leads to IPP production and activation of Vδ2 T cells. The dissociation between the antibody and the N-BP should take place in the lysosomes due to acidic milieu. Eventually, N-BP lead to pAg production and its presentation to Vδ2 T cells. Actually, we cannot exclude that EGFR engagement with the native or N-BP- linked antibody can deliver a signal inside the CRC cells. But all the CRC cell lines tested in our study can proliferate independently of EGFR, because of their compensatory mutations. The inhibitory effect of anti-EGFR on Caco-2 cell proliferation would suggest that this cell line can still depend on EGFR-mediated signaling. This could be explained with the observation that Caco-2, producing the EGFR ligand epiregulin (EREG), can trigger EGFR-mediated autocrine proliferation. If this is the case, the antibody could affect the interaction between EREG and EGFR reducing the proliferative effect. The stronger effect of N-BP-linked antibodies could be explained with the apparently stronger reactivity with the EGFR showed by these ADC compared to the native antibody at least with some CRC cell lines.

  1. Down regulation of EGFR using antibodies upregulates phospho-AXL levels. Authors have to monitor for pEGFR and pAXL levels using western blots.

We have performed the analysis of EGFR-mediated signaling after the engagement of EGFR without or with the native Cet as well as the three different ADCs Cet-IBA, Cet-RIS and Cet-ZA. We have added these data as supplementary figure 5. We analyzed the Caco-2 and HCT-116 cell lines. Overall, we cannot state that the antibody treatment gives to evident effects on the total EGFR, Akt and Erk1 and Erk2 as well as their phosphorylated forms. This could have been expected in the absence of an added ligand. Actually, it appears that pAkt is slightly increased in Caco-2 cell line when treated with Cet-ZA compared to the others antibodies. This can be a compensation mechanism to favor Caco-2 cell survival. Also, this effect was not evident in HCT-116 cell line. The densitometric analysis did not reveal clear differences (> or < than two-fold of basal levels).  These finding would suggest that under our experimental conditions (absence of EGF, not starved cells, culture with medium with FCS) the EGFR signaling is not clearly induced or altered (indeed EGFR p1068 phosphorylation signal was barely detectable). Our main aim was to use the EGFR as a favorable docking molecule for the ADCs, to increase N-BP uptake, causing the activation of the immune system through the generation of small pyrophosphate antigens, as described above.

We also understand the relevance of other receptors such as the AXL receptor tyrosine kinase because it is a quite promising target receptor in cancer therapy. However, we do not have at present the expertise and reagents to study this receptor. Nevertheless, we thank the reviewer for her/his suggestion. Indeed, we will study in the future the possibility of targeting this molecule also using anti-AXL antibodies (such as AF154, DAX-L88 and Tilvestamab (BGB149)) conjugated with N-BPs.

  1. Inhibition of EGFR upregulates low-fidelity polymerases. Authors to monitor the expression of low fidelity polymerases upon treatment with ADCs.

Actually, we did not find a clear induction or inhibition of EGFR signaling, thus it is conceivable that also low fidelity polymerases will be not upregulated. As already reported above, our aim was not to interfere with the EGFR signaling, but use this receptor as a target molecule of the ADC and subsequently trigger Vδ2 T cells. Under our experimental culture conditions (absence of EGF in particular), the EGFR-mediated signaling is barely affected. All the cell lines used in our study have been cultured in vitro, without added EGF ligands, for decades and show typical driving mutations mediating resistance to EGFR-inhibition. Indeed, we chose them also to show that our ADCs are active also against EGFR-signaling independent CRC cells, thus applicable to the majority of CRC cases.

  1. What is the status of EGFR when treated with ADCs? Authors have to check for RNA and protein expression of EGFR.

The surface expression of EGFR is not markedly changed along the period of time of incubation with native cet or N-BPs linked antibodies.

  1. Authors have to discuss the intracellular effects of ADCs when EGFR is used as a target. Authors have to differentiate the molecular mechanism of cetuximab and ADCs.

According to the reviewer’s concern, we have discussed the possible molecular mechanism by which ADC can exert its effects. In particular, this has been added on lines 482-507 of the Discussion section. See the reply to the point 1.

Reviewer 2 Report

Comments and Suggestions for Authors

The manuscript aims to develop and evaluate a novel antibody-drug conjugate (ADC) targeting colorectal cancer cells, specifically focusing on an ADC that combines cetuximab (Cet), targeting the epidermal growth factor receptor (EGFR), with aminobisphosphonates (N-BPs). This ADC is investigated for its potential to kill colorectal cancer cells via activation of γδ T cells.

I have several concerns regarding this study:

1. The use of random lysine conjugation is mentioned, which generally complicates analysis and Chemistry, Manufacturing, and Controls (CMC). Have the authors considered alternative conjugation methods, such as cysteine conjugation or site-specific conjugation, to potentially simplify these aspects?

2. Determining the drug-to-antibody ratio (DAR) using MALDI-MS is unconventional. Techniques such as Hydrophobic Interaction Chromatography (HIC), Reverse Phase HPLC (RP-HPLC), or Quadrupole Time-of-Flight MS (Q-TOF MS) might be more appropriate. It is speculated that the chosen lysine conjugation approach may result in too broad HPLC peaks for effective analysis.

3. The absence of in vivo data significantly hinders the ability to evaluate the promise of this ADC. It is premature to assess its potential without such data.

In conclusion, while the study addresses an innovative approach to treating colorectal cancer, addressing these concerns could significantly strengthen the manuscript.

Author Response

Reviewer 2

The manuscript aims to develop and evaluate a novel antibody-drug conjugate (ADC) targeting colorectal cancer cells, specifically focusing on an ADC that combines cetuximab (Cet), targeting the epidermal growth factor receptor (EGFR), with aminobisphosphonates (N-BPs). This ADC is investigated for its potential to kill colorectal cancer cells via activation of γδ T cells.

I have several concerns regarding this study:

  1. The use of random lysine conjugation is mentioned, which generally complicates analysis and Chemistry, Manufacturing, and Controls (CMC). Have the authors considered alternative conjugation methods, such as cysteine conjugation or site-specific conjugation, to potentially simplify these aspects?

We agree with the concern raised by the reviewer but we performed several attempts to link zoledronate to cysteine instead of using lysine as target amino acid. Indeed, we started trying to prepare the ADC using complete reduction of native interchain disulfide bonds of the mAb followed by alkylation of the resulting 8-cysteine residues with a maleimide-linked payload. Unfortunately, this approach did not give rise to an efficient conjugation of zoledronate. For this reason, we shifted to the conjugation in lysine residues.

  1. Determining the drug-to-antibody ratio (DAR) using MALDI-MS is unconventional. Techniques such as Hydrophobic Interaction Chromatography (HIC), Reverse Phase HPLC (RP-HPLC), or Quadrupole Time-of-Flight MS (Q-TOF MS) might be more appropriate. It is speculated that the chosen lysine conjugation approach may result in too broad HPLC peaks for effective analysis.

We partially agree with this concern. Indeed, ADC in which the drug is linked to cysteine residues should be analyzed through the techniques listed by the reviewer. Also, linking to lysine could be performed with these techniques. However, when drugs are linked with the antibody through lysine residues, the MALDI-MS can be used with appropriate results. This has been reported in the literature: 

  • Giannini G, Milazzo FM, Battistuzzi G, Rosi A, Anastasi AM, Petronzelli F, Albertoni C, Tei L, Leone L, Salvini L, De Santis R. Synthesis and preliminary in vitro evaluation of DOTA-Tenatumomab conjugates for theranostic applications in tenascin expressing tumors. Bioorg Med Chem. 2019 Aug 1;27(15):3248-3253. doi: 10.1016/j.bmc.2019.05.047. Epub 2019 May 31. PMID: 31208798.
  • Milazzo FM, Vesci L, Anastasi AM, Chiapparino C, Rosi A, Giannini G, Taddei M, Cini E, Faltoni V, Petricci E, Battistuzzi G, Salvini L, Carollo V, De Santis R. ErbB2 Targeted Epigenetic Modulation: Anti-tumor Efficacy of the ADC Trastuzumab-HDACi ST8176AA1. Front Oncol. 2020 Jan 23;9:1534. doi: 10.3389/fonc.2019.01534. PMID: 32039017; PMCID: PMC6989603.
  • Källsten M, Hartmann R , Artemenko K , Lind SB , Lehmann F , Bergquist J . Qualitative analysis of antibody-drug conjugates (ADCs): an experimental comparison of analytical techniques of cysteine-linked ADCs. 2018 Nov 5;143(22):5487-5496. doi: 10.1039/c8an01178h. PMID: 30289422.
  • Abedi M, Ahangari Cohan R, Mahboudi F, Shafiee Ardestani M, Davami F. MALDI-MS: a Rapid and Reliable Method for Drug-to-Antibody Ratio Determination of Antibody-Drug Conjugates. Iran Biomed J. 2019 Nov;23(6):395-403. doi: 10.29252/ibj.23.6.395. Epub 2019 May 20. PMID: 31104399; PMCID: PMC6800535.

This is the reason we have chosen MALDI-MS instead of the others.

  1. The absence of in vivo data significantly hinders the ability to evaluate the promise of this ADC. It is premature to assess its potential without such data.

We agree with the reviewer’s concern. Considering this relevant point raised, we have inserted in the discussion section that the lack of a direct demonstration of efficacy of these antibodies in vivo is a limitation of this study (lines 554-556).  

However, we want to better explain what we wanted to state in this manuscript. Actually, our experiments were aimed to point out the possibility to link different aminobisphosphonates (N-BPs) with the anti-EGFR antibody cetuximab to demonstrate the major efficacy of one or another N-BPs. We perfectly understand that our data do not show the efficacy in vivo of our ADC. However, the Vδ2 T cells activated by zoledronate processed in targeted tumor cells with the antibody cetuximab are not so frequent in mouse and do not respond to small pyrophosphates as this happens in humans. Thus, the use of murine models should imply xenogenic transplantation of human tumor cells and effector cells. The subcutaneously xenogenic murine models do not resemble the human situation of response to CRC. Also, the setting of an orthotopic murine model with human cells is not easily set up in our lab. The same could be stated for the generation of humanized mice, where a strong, specific expertise is needed. In any case, we have to get the permission for animal studies (3-6months) to our Minister of Health and at least further three months would be needed to get an answer. This indicates that to reply reviewer’s query at least one-year work is necessary. Actually, at present we are waiting for the permission of toxicity tests in mice, but we did not receive the permission yet. After these experiments we are going to assess the efficacy of the antibodies in mice to have the proof that indeed these antibodies could be used in a clinical setting.

Round 2

Reviewer 1 Report

Comments and Suggestions for Authors

Major Comments: 

The authors have addressed the comments verbally. It would be great if authors would include these in the discussion or introduction. 

Author Response

The authors have addressed the comments verbally. It would be great if authors would include these in the discussion or introduction.

Following the reviewer’s comment, we inserted the replies of the first round of reviewing in the discussion section. These new additions have been highlighted in blue to distinguish them from the previous additions in red.

In particular, to the first and last comment of the previous round, we have already inserted the method for WB analysis from lines 189-208. In addition, we added in the discussion section the reply to the first point and the last one from lines 488-513 (see first revised version). These additions are in red because they correspond to the previous revised version.

Also, in this last revised version, we added some comments limited to reviewer’s reply, regarding the other points of the previous reviewing always in the discussion section lines 489-498 and 518-526 (see blue additions). By adding these considerations, we also added some new references (new ref 35-39).

Reviewer 2 Report

Comments and Suggestions for Authors

Accept in present form

Author Response

Thank you for your positive evaluation.

Round 3

Reviewer 1 Report

Comments and Suggestions for Authors

Authors have addressed thee comments.